# Role of Transglutaminase 2 in Migration of Tumor Cells and How Mouse Models Fit

**DOI:** 10.3390/medsci6030070

**Published:** 2018-08-30

**Authors:** Ajna Bihorac

**Affiliations:** Department of Genetics and Bioengineering, Yeditepe University, 34755 Istanbul, Turkey; ajnabihorac@gmail.com

**Keywords:** tumor cell migration, TG2, mouse models

## Abstract

A search for the “magic bullet”, a molecule, the targeting abilities of which could stop the migration of tumor cells, is currently underway, but remains in the early stages. There are still many unknowns regarding the cell migration. The main approach is the employment of mouse models, that are sources of valuable information, but still cannot answer all of the questions. One of the molecules of interest is Transglutaminase 2 (TG2). It is a well-described molecule involved in numerous pathways and elevated in metastatic tumors. The question remains whether mice and humans can give the same answer considering TG2.

## 1. Introduction

Tumors consist of a heterogeneous population of cells, and the heterogeneity is the consequence of genomic instabilities such as chromosomal rearrangements, DNA mutations, and epigenetic changes [1]. Malignant tumors can invade the surrounding tissue and travel to distant parts of the body where they develop into secondary tumors (macrometastases) [2]. It has been known for some time that metastases from certain tumors develop in certain organs. In 1889, English surgeon Stephen Paget suggested that some tumor cells (“seeds”) have a specific attraction towards certain organs (“soil”); this is also known as the “seed and soil” hypothesis [3]. Ewing had a different opinion, and in 1928, he suggested that the anatomy and the vasculature control the metastases found in clinical practice [4]. Since both theories survived over time, the conclusion is that both ways can contribute to the affinity of cancer cells towards certain organs [5]. One question of interest is when do tumor cells gain the ability to metastasize. The main opinion is that the extracellular environment can influence the metastatic potential since cells have to respond to immune cells attack, have restricted accessibility of nutrients and oxygen and respond to changes of extracellular matrix (ECM) components [6]. Additionally, by secreting various growth factors, cells around the neoplasm fibroblasts, epithelial cells, and endothelial cells also play roles in the activation of the tumor promoting pathways [7].

### 1.1. Migration of the Tumor Cells

There are two main forms of cell migration: single cell and collective cell migration. Recently, it has been suggested that the migration of cancer cells is more plastic, thus tumor cells can change their methods of movement from one form to another, depending on external signals and intracellular regulation [8]. A developmental regulatory program, epithelial–mesenchymal transition (EMT), has been suggested as a way for transformed epithelial cells to become migratory, invasive, and to have the ability to metastasize [9,10]. Another single cell form of migration is ameboid cell migration which describes individual cells that slide through the existing interstices in the ECM [11]. On the other hand, in collective cell invasion, cancer cells make small nodules or strands during metastasis to the secondary organs [11,12,13].

At the biochemical level, EMT depends on signaling pathways, leading to the up-regulation and down-regulation of biomarkers responsible for the phenotype of mesenchymal and epithelial cells by the transcription factors Snail, Twist, Slug/Snail2, Zeb1, and Zeb2/SIP1 [14,15,16]. Epithelial cadherin (E-cadherin) and integrins α6β1 and α6β4, markers of epithelial cells are substituted with the neural cadherin (N-cadherin) and fibronectin binding integrins α5β1 and αVβ3, markers of mesenchymal cells. Additionally, the invasion of tumor cells is enabled by the increased expression of matrix metalloproteinases (MMPs), such as MMP2 and MMP9 [17]. The oncogenic EMT usually generates poorly differentiated cells in vivo with a hybrid, “metastable” phenotype having the features of both, epithelial and mesenchymal cell type [18,19]. The steps in the oncogenic EMT are less ordered and lack coordination [20,21], because the cancer cells contain numerous genetic changes and receive constant signals from the pro-tumorigenic microenvironment [15].

The migration can be considered collective if cells migrating keep contact and influence one another while moving [22]. The collective migration, in the context of cancer is called collective invasion [23]. The activity of MMPs is an important factor in collective migration as shown by grafted Madin–Darby Canine Kidney (MDCK) cells in vivo, in which the elevated expression of MMPs is sufficient to trigger tissue invasion [24]. Additionally, in two-dimensional (2D) space collective migration can be seen as sheet movement, a characteristic migration for metastable phenotype, thus suggesting activation of EMT. Besides the EMT program, MMPs can be provided by stromal fibroblasts surrounding the neoplasm, thus allowing the invasion [22]. In metastatic inflammatory breast cancer, tumor emboli or clusters maintaining p120 and E-cadherin expression were shown to be responsible for the progression of the disease, suggesting collective invasion [25]. On the other hand, the presence of E-cadherin does not facilitate the migration in 2D space [22]. Recently, it was shown that the decision to migrate as a single cell or as a collective is dependent on transforming growth factor beta (TGFβ) signaling, which is also known as a key regulator of EMT. Neoplastic cells expressing transforming growth factor beta receptor 2 (TGFβ RII) migrate predominantly as single cells or strands at the tumor–stroma border and express markers of EMT, while cells lacking the receptor show largely collective migration as large cell clusters with no evidence of EMT [25,26].

On the other hand, in the invasive front of melanoma, TGFβ influences amoeboid migration, since it promotes cell rounding, membrane blebbing, contractility and invasion [13]. Melanoma cells from the primary tumor mostly spread through the lymphatic vasculature, while distant organ dissemination requires cells to enter the blood circulation [27]. Melanoma arises from melanocytes that are derived from the neural crest and go through EMT during development [13]. It was shown, in general, that metastasis via the hematogenous system is alleviated by EMT, while the cells metastasizing via the lymphatic system can be carried to the lymph nodes with lymphatic fluid [28]. Tumor xenograft studies of breast cancer have shown cancer cells moving in invasive fronts using the amoeboid form of migration, as well [29]. The mesenchymal–amoeboid transition (MAT) connects mesenchymal-like migration with a motility similar to that of the amoeba *Dictyostelium*, which is characteristic of membrane blebs and has no need for MMPs [30]. The main factors responsible for the cytoskeleton reorganization that influences any type of cell motility are small GTPases of the Rho family. Rac and Cdc42 GTPases are the most important for mesenchymal type of migration and the formation of lamellipodia and filopodia, whilst amoeboid migration depends on Rho GTPase and actomyosin contraction leading to membrane blebbing. [8]. One of the factors influencing neoplastic cell migration is ECM. If ECM is rigid, MMPs will be necessary for migration [30]. For example, the invasion of MDA-MB-231 and HT-1080 multicellular spheroids in vitro is dependent on collagens being cross-linked or not. Amoeboid migration can only be detected in non-crosslinked collagen gels, while in gels consisting of covalently crosslinked collagen I, or in human mammary gland explants, migration is fully dependent on the activity of MMP14. From this it can be concluded that amoeboid migration might be important in the metastasis of cells to tissues missing an interstitial collagen network (e.g., the brain), or in cases where neoplastic cells are using already existing ECM tunnels made by tumor-infiltrating fibroblasts, smooth muscle cells, or endothelial cells [31]. Additionally, when primary melanoma explants fixed into 3D collagen matrices were treated with β1 integrin neutralizing antibodies, this changed the migration from protease-dependent collective to protease-independent, single cell amoeboid migration [32]. It is noteworthy that the main tissue type in the chordate embryo is epithelial; thus, the mesenchyme is derived from the epithelium and can convert back to the epithelial phenotype [9,33].

### 1.2. Metastasis in Mice

Epithelial–mesenchymal transition in mouse tumorigenesis was described for the first time over 100 years ago, using the term “carcinosarcoma” [34,35]. EMT tumors have been described as “spindle-cell” tumors, containing cytokeratins, vimentin, and α-smooth muscle actin (SMA), and lacking E-cadherin [36]. Additionally, Snail1 up-regulation and the p53 mutation have been found in EMT mouse mammary tumors [37,38]. In the doxycycline-inducible bitransgenic mouse model for HER2/neu-induced mammary carcinogenesis (MTB/TAN) [39], spontaneous mammary recurrent tumors showed the up-regulation of Snail and exhibited characteristics of EMT, with the loss of E-cadherin [38]. Additionally, the p53 null mammary transplant mouse model [40], besides basal-like and luminal subtypes, was also shown to form claudin-low tumors expressing the core EMT gene expression signature [37].

In human tumors, EMT has been associated with metastasis, drug resistance, and tumor recurrence [38]. For example, Snail1 has been used as a biomarker of metastasis incidences in human breast carcinomas [41], and the loss of E-cadherin is considered to be a poor prognostic sign that is connected with the change from benign to malignant tumors [42]. On the other hand, mouse mammary Met-1 AS-OPN (osteopontin) [43] tumors show up-regulated Snail1 and a loss of metastatic potential. Met-1 AS-OPN tumors consist of spindle cells and are positive for mesenchymal markers such as vimentin, with no detected epithelial markers, which implies that these tumors occur as a result of EMT of Met-1 epithelial cells [43,44]. The EMT tumors of mouse mammary glands are locally aggressive and invade local tissue, but so far, no single distant metastasis has been observed from spontaneously-formed neoplasms [35,44]. It may be that the spindle cell phenotype is a biological dead-end for mice, and that these cells lose the necessary plasticity to return to the epithelial phenotype required for the progression of micrometastasis to macrometastasis [35]. On the contrary, metastatic cells from mammary metastatic mouse models have been observed to travel to the lungs as clusters of epithelial cells via the hematogenous system [45]. In the MCH66 metastatic murine mammary tumor model, metastasis is reliant not on the degree of invasiveness, but on cancer angiogenesis. The tumor grows as a poorly differentiated adenocarcinoma with fibrous stroma and no signs of invasion. The primary neoplasm consists of cell groups completely surrounded by blood vessels developed in dilated sinusoidal structures; thus, the tumor islands are encircled with blood. Metastasis of this model is dependent on intravasation of the neoplastic groups and their transport to the target organs, where a tumor embolus still covered by the endothelial cells is mechanically detained in an arteriole and the tumor cells keep on proliferating [46].

There is a possibility that the EMT-phenotype may be misleading as a marker for metastases in murine models. The 67NR cells show features of EMT, including the spindle phenotype and a loss of E-cadherin, but they are non-metastatic in vivo, which could be proof that the spindle phenotype is a biological dead-end for mice. In support of this finding, highly metastatic mouse breast cancer 4T1 cells are epithelial-like with all of the characteristics of epithelial cells, such as the presence of E-cadherin and the lack of N-cadherin. On the contrary, the metastatic 66cl4 cell line isolated from the same type of tumor as the 4T1 cells and characterized with high expression of N-cadherin and the loss of E-cadherin is almost non-migratory and does not express invasive properties in vitro. All of the above factors suggest that migratory and invasive properties in vitro are not connected with EMT markers and metastatic potential in vivo for the mouse breast cancer models [47]. Besides, in the mouse model of renal cell carcinoma (RCC; performed with the Renca cell line) increased metastatic spread to the lungs is influenced by the genetic changes (knockout of Von Hippel–Lindau, VHL, the tumor suppressor gene) that led to the EMT phenotype in vitro [48]. This indicates that EMT may not be necessary for breast cancer metastasis in mice, but it may influence the metastatic aggressiveness of the mouse RCC model. All of the above results suggest that metastasis in mice may depend more on a metastable state than the classical mesenchymal one.

Intravital imaging of B16F2 melanoma cell migration in mice showed the heterogenous behavior of the cells. Some melanoma cells migrated as single cells, while others followed. Single cells used an amoeboid type of motility, while the cells following the same path used multicellular streaming [49]. Additionally, Ca^2+^ is a significant regulator of cell migration, and transitory Ca^2+^ release from the intracellular depos is necessary to induce membrane blebs in D54 glioma cells isolated from BALB/c Scid mice in situ. Bleb formation can be disturbed by affecting intracellular Ca^2+^ homeostasis [50]. Moreover, murine 4T1-Luc mammary tumor cells expressing recombinant luciferase were injected intravenously in BALB/cByJRj mice, and they metastasized to the lungs, as detected with bioluminescent measurement. Treatment with chondramide, an actin targeting compound, decreased the metastasis [51].

## 2. Transglutaminase 2

Transglutaminases (TGs) are the family of enzymes that catalyze covalent bond formation between free amino groups (e.g., protein or peptide-bound lysine) and the γ-carboxamide groups of peptide-bound glutamine [52]. In 1957, Waelsch and colleagues showed that the soluble liver protein fraction can incorporate labelled amines into proteins in the presence of Ca^2+^ (Figure 1), and tissue transglutaminase (tTG), also known as transglutaminase 2 (TG2) was identified [53]. Besides the gene for TG2 (TGM2), eight more members have been identified (TGM1, TGM3, TGM4, TGM5, TGM6, TGM7, F13A1, and EPB42) in mammals so far, coding different enzymes of transglutaminase family [52].

Tissue transglutaminase is present in the cell’s cytosol, mitochondria, and nucleus, and it can be found on the cell surface and in the ECM with a ubiquitous presence throughout the body [54,55]. Besides its primary enzymatic activity of calcium-dependent post-translational protein modification, it can bind GTP and ATP, acts as a G protein, has protein disulfide isomerase activity, acts as a protein kinase, and has isopeptidase activity [56,57]. It has been contemplated that expressing, at minimum, one other catalytically active TG besides TG2, as was found in nearly all cell types in the body, gives suitable grounds for tissue transglutaminase to develop its additional features [58].

Activated tissue transglutaminase introduces intermolecular isopeptide bonds and thus modifies ECM proteins. The bonds have an influence on the stability of the extracellular matrix and cell adhesion/spreading. Substrates of TG2 include fibronectin, collagen, fibrinogen, fibrin, laminin/nidogen, osteopontin, and vitronectin [59]. These bonds are stable, covalent, and resistant to proteolysis, and so are resistant to chemical, enzymatic, and physical disruption [60]. Additionally, TG2 can introduce intramolecular bonds, thereby changing the conformation stability and biological function of the proteins (e.g., inhibitor of the nuclear factor kappa-light-chain-enhancer of activated B cells (NF-κB) pathway, IκBα) [61]. Besides crosslinking, TG2 may incorporate primary amines into cytoplasmic proteins, thus altering their activity [52]. For example, TG2-moderated serotonylation of RhoA is connected with human pulmonary hypertension, involving pulmonary artery smooth muscle cells [62]. In addition, serotonylation of α-actin and other proteins in the contractile apparatus of vascular smooth muscle cells increase arterial isometric contraction [63]. Furthermore, the serotonylation that activates Rac1 signaling in cortical neurons is dependent on TG2 [64]. On the contrary, if there are no amine substrates, water serves as a nucleophile and so, glutamine residues can be deaminated by TG2 to glutamic acid residues, influencing the conformation, activity, and interactions of target proteins. The deamidating function is important in the deamidation of peptides from the wheat protein gliadin, which activates the immune response connected to celiac disease [65,66].

### 2.1. Transglutaminase 2 in Cancer

Elevated expression of TG2 has been found in secondary metastatic tumors of glioblastoma [67], pancreatic carcinoma [68], RCC [69], breast carcinoma [70], lung carcinoma [71], melanoma [72], and ovarian carcinoma [73]. In addition, a study of 30,000 genes from samples of cancers showed that TG2 is one of the genes that is overexpressed in pancreatic cancer [74], and one of eleven proteins that is increased in metastatic lung carcinoma [75]. Elevated expression of TG2 may be connected with epigenetic alterations, as in doxorubicin-resistant breast tumor cells that have high levels of TG2 and hypomethylated TGM2 promoter, while doxorubicin-sensitive cells have a hypermethylated promoter [76]. Additionally, the TG2 gene has regulatory elements that respond to cytokines, including TGF-β, interleukin 6 (IL-6) and tumor necrosis factor alpha (TNF-α) [59], suggesting that the tumor microenvironment may also have a role in increased TG2 expression. For example, TGF-β1 elevates TG2 expression, while by binding to TGF-β1 on the cell surface, TG2 induces its conversion to the biologically active form, thus forming a positive feedback loop [77].

Transglutaminase 2 on the cell surface is involved in cell adhesion via its role as an integrin-binding coreceptor for fibronectin and a stabilizer of ECM [78,79]. Cell-surface TG2 interacts with β1 and β3 integrins and supports their connection with the gelatin-binding domain of fibronectin that does not have intrinsic integrin binding sites. TG2 also has some affinity for heparin, a highly sulphated analogue of heparan sulphate which is abundantly present in the cell surface/ECM. Additionally, the heparan sulphate proteoglycan, syndecan-4, is another binding partner for extracellular TG2 [80].

In secondary metastatic and chemotherapy resistant tumors, TG2 is often up-regulated, whereas during tumor development and in primary tumors, TG2 is down-regulated [81]. The decrease in TG2 expression in primary tumors may be achieved by epigenetic gene silencing, since the hypermethylation of CpG islands partly covers both the transcriptional and translational start sites of the TGM2 gene in cultured human breast tumor cells [76]. Since TG2 acts as a stabilizer of ECM, which has anti-angiogenic properties and inhibits malignant cell proliferation and migration, a decrease in TG2 expression and activity in primary tumors may allow tumor spread and facilitate angiogenesis [82]. The survival of cancer cells that break out into the blood vessels is dependent on their ability to dock and adhere at a distant site, which may explain the increased expression of TG2 in metastatic tumors considering its role in cell adhesion [83]. Additionally, TG2 up-regulation in drug resistant tumors may be connected with its link to a number of survival pathways, since TG2 interaction with integrins and fibronectin on the cell surface may lead to the activation of cell survival and anti-apoptotic signaling pathways [84,85]. Additionally, the enzyme might covalently incorporate cytotoxic drugs, thus completely abrogating their activity. Numerous anticancer agents, such as bleomycin, adriamycin, mithramycin and actinomycin D, are considered to be amine substrates for transglutaminases [82,86]. Additionally, the TG2/NF-κB/HIF1α (hypoxia-inducible factor 1-alpha) pathway appears to be necessary for elevated expression of the EMT transcription factors Twist, Snail, Zeb1, and Zeb2 in MCF10A epithelial cells expressing TG2. The interaction of TG2 with the p65/p50 complex in the cytoplasm and the promotion of IκBα degradation via the proteasomal-independent pathway, allows the translocation of the complex to the nucleus. The complex is then moved to the NF-κB binding site in the HIF1α promoter, resulting in its increased expression and further up-regulation of the transcription factors Twist, Snail, Zeb1, and Zeb2. It is possible that TG2’s interaction with NF-κB helps recruit co-activators or repressors to the promoter site, and TG2 is involved in p65 subunit phosphorylation [87]. Additionally, the up-regulation of TG2 in MCF10A cells also increases the number of cells expressing CD44^high^/CD24^-/low^ stem cells markers, while influences down-regulation of an epithelial marker CD326 [88]. Similarly, in human ovarian tumor cells, SKOV3, the cell population expressing CD44^+^/CD117^+^ stem cell markers is significantly reduced by the knock-down of TG2 and proposed mechanism is dependent on TGFβ signaling [89].

### 2.2. Transglutaminase 2 in Mouse Models

The crosslinking activity of TG2 influences ECM deposition and angiogenesis in mouse models, as shown after intratumor injection of active TG2 into mice with CT26 colon carcinoma tumors [90]. Furthermore, B16-F1, B16-F6, and B16-F10 mouse melanoma tumor progression and metastasis were increased in TG2 knockout mice compared to wild type (wt) controls [91]. TG2 knockout mice have decreased NFkB activation after lipopolysaccharide-induced septic shock. Additionally, these mice are less prone to kidney damage compared to controls, because of reduced infiltration of macrophages and myofibroblasts, decreased collagen synthesis, and reduced TGF-β activation [77,92]. On the other hand, TG2^−/−^ mice have normal collagen crosslinking and are not protected from liver fibrosis, suggesting that the lysyl oxidase (LOX) family has a role in disease progression [93]. Although TG2^−/−^ mice have shown that there is no role for TG2 in glucose homeostasis, at least in mice [94], in diabetic mice, hyperglycemia influences the activation of TG2 via vascular endothelial growth factor (VEGF) in endothelial cells as well as dissociation of beta-catenin from vascular E-cadherin and subsequent adherens junction dissolution, along with the stress fiber formation [95], thus showing the role of TG2 in endothelial–mesenchymal transition (EndoMT) in diabetic mice. Furthermore, TG null mice show a serious defect in ATP production [57,96], and embryonic fibroblasts from TG2^−/−^ mice have fragmented mitochondria with altered morphology, depolarization of the mitochondrial membrane, and reduced reactive oxygen species (ROS) formation [57,97,98]. Moreover, TG2 null mice develop autoimmune disease when more than 1 year of age [99,100].

## 3. Conclusions

The main question that we need to consider is whether mouse models can mimic all aspects of a human disease. The physiological differences between mice and human are numerous, including differences in lifespan, structure of the stromal elements [101], and the presence of telomerase [102]. Additionally, in mice, the metabolism of xenobiotics is different [103], along with mutation rates [104] and transformation ability [5,105,106]. In mouse models, the occurrence of metastasis is frequently much lower than the presence of primary tumors, and organ tropism in mice is often different than in humans. It is also important to mention that, as in the mouse skin tumor model, an analysis of metastasis usually cannot be performed because the animals are killed when the primary tumors grow to their maximum size; thus, visible metastases are not detected [107]. Because “mice are not human”, there will be differences in the outcomes of the same event between the two, so there is still a need to develop an ideal mouse model that includes every characteristic of human metastatic disease [5,35]. Considering the above, and the search for the “magic bullet” in fighting metastatic tumors, TG2 may be a key regulator in the cell migration responsible for the dissemination of neoplastic cells. TG2 knockout mice have been shown to have increased B16-F1, B16-F6, and B16-F10 melanoma tumor progression compering to wild type. In addition, activation of NFkB and TGFβ are decreased. TG2-deficient mice have altered morphology of mitochondria and defect in ATP production. Furthermore, in diabetic mouse models, TG2 may be involved in endothelial–mesenchymal transition (EndoMT). In humans, the stress situations leading to elevated levels of extracellular TG2, induce EndoMT in endothelial cells by regulating TGFβ1 signaling [108]. Moreover, TG2 is included in the activation of other pathways, such as NF-κB, HIF1α, etc., all of which are responsible for cancer progression. Additionally, via its cross-linking activity it influences the ECM surrounding the neoplasm. Since one of the substrates is collagen, by fine tuning of TG2 activity, mesenchymal and amoeboid migration may be controlled. Moreover, TG2 influences the Rho family of GTPases, as mentioned above. It may be worth considering whether TG2 can influence the migration of the cells via its GTPase role as well. Recently, it was shown that 2D migration of epidermal cancer stem cells (ECS) is significantly reduced by transamidase site-specific inhibitors (NC9, VA4, VA5 and CP4d), that by interacting with the transamidase catalytic center produce a conformational change thus inactivate TG2 GTP binding site, necessary for cancer stem cell survival and migration [109,110] Additionally, in D54 glioma cells, the formation of blebs is dependent on Ca^2+^. Mitochondrial Ca^2+^ homeostasis is of great importance for the migration of cells [111]. After stimulation of the rat insulinoma cell line (INS-1E) with glucose, mitochondrial proteins involved in Ca^2+^ homeostasis, such as the voltage-dependent anion-selective channel (VDAC) protein, prohibitin, and different ATP synthase subunits, were discovered to be substrates of TG2 [57,112], indicating that TG2 activity is important in this process too. Since all research must have a starting point, mouse models along with cell culture experiments may give certain answers for the development of treatments, at least until we develop the perfect model. As written by William Arthur Ward, “The pessimist complains about the wind; the optimist expects it to change; the realist adjusts the sails”.

## Figures and Tables

**Figure 1 medsci-06-00070-f001:**
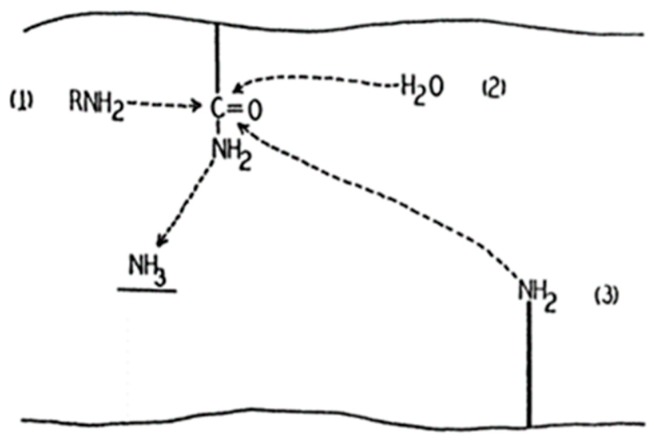
First draft of the transglutaminase catalyzing reaction [53].

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
