# Peer review of "Role of Transglutaminase 2 in Migration of Tumor Cells and How Mouse Models Fit"

_medsci, 2018, doi:10.3390/medsci6030070_

Round 1

Reviewer 1 Report

The review authored by Dr. Ajna Bihorac summarizes the involvement of tissue-transglutaminase in cancer cells migration and how the mouse model s are useful in the study of this phenomenon.

T he review is quite exhaustive, well written and the citations reported are sufficient to cover most of the knowledge about the topic. My general opinion about this work is positive, I believe that it could be better suitable for publication on Medical Sciences after some minor revisions.

Minor revisions:

1.lane 19: this sentence should be improved (too many repetition of “tumors”).

2. Cancer stem cells represent a new and very intriguing target for therapy. The author could mention and elaborate this aspect (just as an example: Fisher et al., Cancer Res 2016;76(24):7265-7276; Kerr et al., Oncogene 2017;36(21):2981-2990).

Author Response

    Thank you for your review. I have corrected and added couple of sentences about cancer stem cells. Thank you for pointing me to the paper Kerr et.al., 2017.

Line 19 was corrected as:  Malignant tumors can invade the surrounding tissue and travel to distant parts of the body where they develop into secondary tumors (macrometastases) [2].  

From line 240 I added : Additionally, the up-regulation of TG2 in MCF10A cells also increases the number of cells expressing CD44high/CD24-/low stem cells markers, while influences down-regulation of an epithelial marker CD326 [88]. Similarly, in human ovarian tumor cells, SKOV3, cell population expressing CD44+/CD117+ stem cell markers is significantly reduced by the knock-down of TG2 and proposed mechanism is dependent on TGFβ signalling [89]. 

and in line 290: Recently, it was shown that 2D migration of epidermal cancer stem cells (ECS) is significantly reduced by transamidase site-specific inhibitors (NC9, VA4, VA5 and CP4d) that by interacting with the transamidase catalytic centre produce a conformational change thus inactivate TG2 GTP binding site, necessary for cancer stem cell survival and migration [109, 110]. 

Reviewer 2 Report

Nice work!

This manuscript “Role of TG2 in migration of tumor cells and how mouse models fit” describes the differences of epithelial–mesenchymal transition (EMT) in mouse and human. By focusing on TG2 activities, author (Dr. Bihorac) summarized clearly what have been reported from the mouse model with TG2 and raised a question that can we apply the findings of TG2 from mouse model to fit in clinical application? TG2 is a multi-faceted protein that plays controversial roles in normal and tumor cells, furthermore, TG2 can be upregulated or downregulated concordance with microenvironment alteration. Target TG2 to stop the migration of tumor cells only in its early phase of investigation, this article provides a deep and inspired review for the field of TG2 study.

Only the following mistakes needed to be corrected.

Line 86, to blebbing. [8].?

Line 275, compering to wt. Define wt?

Author Response

    Thank you for your review. I have corrected all mentioned.

In line 86 I corrected as to membrane blebbing. [8].

In line 275 I wrote to wild type.